# How does a change in climate variability impact the Greenland ice-sheet surface mass balance?

Tobias Zolles[1,2] and Andreas Born[1,2]

[1]Department for Earth science, University of Bergen, Bergen, Norway
[2]Bjerknes Centre for Climate Research, Bergen, Norway

**Abstract.** Given the long response time of ice sheets, simulations of the Greenland ice sheet typically exceed the availability of input climate data to reliably simulate the fast processes underlying surface mass balance. Strong feedback processes are known to make the mass balance sensitive to inter- and intra-annual variability. Even simulations with climate models do not always cover the full period of interest, motivating bridging these gaps using relatively coarsely resolved climate reconstructions or temporal interpolation methods. However, both of these approaches usually only provide information about the climatological average but not variability. We investigate how this simplification impacts the surface mass balance using the Bergen Snow Simulator. The model was run for up to 500 years using the same atmospheric climatology, but different synthetic variabilities.

While changing inter-annual variations has an impact of less than 5% on the surface mass balance of the Greenland ice sheet, neglecting intra-annual variability by using a daily climatology causes a 40% change in mass balance. Decomposing the total effect into contributions from different input variables, the biggest contributor is precipitation followed by temperature. Using a daily climatology, a small amount of snowfall every day overestimates the albedo and thus SMB. We propose a correction that re-captures the effect of intermittent precipitation, reducing the SMB overestimation to 15-25%. We conclude that simulations of the Greenland surface mass and energy balance should be forced with a transient climate, in particular for models that are calibrated with transient data.

*Copyright statement.* TEXT

## 1   Introduction

The Greenland ice sheet is one of the main contributors to sea-level rise. Future projections show that uncertainty associated with the atmospheric climate forcing becomes dominant within the next century (Aschwanden et al., 2019). Yet, simulations of the Greenland ice sheet are subject to many sources of uncertainty. The climate forcing itself is inherently uncertain as a result of uncertain emission scenarios (O'Neill et al., 2016), but it also depends on the used climate model, which remains a major source of uncertainty until the end of the century (Holube et al., 2021). Furthermore, ice-sheet models may be forced with a multiyear climatology, monthly averages or daily data with unclear consequences due to the non-linearity of the SMB (Mikkelsen et al., 2018).

Paleo simulations of ice-sheets are often based on proxy temperature reconstructions (Van de Berg et al., 2008; Robinson et al., 2011). Because proxy data has a limited temporal resolution, it is often impossible to accurately reconstruct inter- and intra-annual variability. While it is common practice to use a temperature index to interpolate between the coldest (Last Glacial Maximum) and the warmest (Present Day) state (e.q. Forsström and Greve, 2004; Alvarez Solas et al., 2018), it has not been studied what impact additional variability on unresolved shorter time scales would have. Proxies vary greatly in their temporal resolution, so we investigate the variability on multiple time scales (50 - 500 years). Although the initial question arises from proxy and climate reconstruction it is equally applicable to projections of the distant future of the Greenland ice sheet.

In this study, we perform simulations using the latest version of the BErgen Snow SImulator (BESSI) (Zolles and Born, 2021). The model was calibrated using GRACE satellite data (Wiese et al., 2016) and RACMO simulations (Noël et al., 2018; Fettweis et al., 2020a; Holube et al., 2021). The model is designed for the simulations of long time scales, leading to a trade off between complexity and computational efficiency. Therefore, we need a representative climate forcing for longer time periods.

Input data to force BESSI is derived from the ERA-interim reanalysis data set, instead of using an artificial inter-annual variability or internal climate model variability (Semenov, 2008; Verdin et al., 2018). Firstly, the rapidly increasing temperature over the last 50 years is a good example of a non-representative climatological average. Secondly, ERA-interim provides a reasonable natural variability and daily data is available over the entire Greenland Ice-sheet at a sufficiently high spatial resolution (Berrisford et al., 2011). Potential climate model data for climate reconstructions and projections will be of a similar or lower resolution. We create climate variability of different time scales by reordering individual full years of the ERA-interim record. For a longer simulation duration the ERA-interim period is copied multiple times. We use ERA-interim as its resolution is of the same order of magnitude as most General Circulation climate Models (GCM) and refrain from using data of higher resolution models such as MAR (Fettweis et al., 2017) or RACMO (van Meijgaard et al., 2008) as those will not be available for the most of the past i.e. the last glacial cycle. We choose the current rapid climate change as it provides an upper uncertainty estimate for the entire glacial. Furthermore, the model sensitivity of the surface mass balance model has been evaluated previously for this time period (Zolles and Born, 2021).

This leaves us with three goals of the study:

- Quantify the uncertainty associated with inter-annual variability and climatological forcing

- Identify the reasons for and potentially reduce this uncertainty

- Find a procedure to create a representative climate forcing for the past based on temperature proxies

In section 2 we will give a brief description of the surface mass balance model and the set-up of the climate ensemble used in this study. The results in section 3 are split into the uncertainty of inter-annual variability, individual forcing variables, and precipitation and associated albedo impact. After that, we discuss our findings in section 4 and conclude in section 5.

## 2 Model setup

### 2.1 Snow model - BESSI

The study uses the Bergen Snow SImulator (BESSI), which calculates the mass and energy balance with a daily time step (Born et al., 2019). It compares well to other surface mass balance models over Greenland with a slight positive bias for melt regions (Fettweis et al., 2020a). The model version used here is described in detail in Zolles and Born (2021) so that we will only provide an abridged description here. The model domain is based on a stereographic projection of Greenland and uses an equidistant grid with a resolution of 10 km. The model uses a mass based vertical grid of 15 layers, with up to 500 $\text{kg m}^{-2}$. The model uses five input fields with a daily resolution: surface temperature, total precipitation, dew point, and down-welling long- and shortwave radiation. A full energy balance is calculated at the surface including diffusion of heat in the snow pack and latent contributions from freezing and melting of water and liquid precipitation. Liquid water in the snow is explicitly represented. Mass changes due to melting, precipitation, or sublimation processes. The model parameters have been tuned using a multi-variate calibration towards RACMO (Noël et al., 2018) and the GRACE data set (Wiese et al., 2016).

### 2.2 Atmospheric climate forcing

As forcing, we use the daily ERA-interim reanalysis data from 1979-2017 (Uppala et al., 2011). The input variables of atmospheric temperature, precipitation, dew point, and short and long-wave radiation are bi-linearly interpolated to a 10x10 km grid over Greenland. This initial forcing data of 39 years is then taken 12 times to represent longer time periods. We define the natural transient forcing as the ERA-interim forcing in the true historical order and then looping forward and backward (F-BWD), as explained below.

We arrange the original transient forcing in four different ways: repeating the ERA-interim forcing in its original order multiple times (forward, FWD), repeating the same data in reverse order (backward, BWD), alternating between FWD and BWD to avoid the abrupt transition between the forcing years 2017 and 1979 (forward-backward, F-BWD), and again the same in reverse (backward-forward, B-FWD). This already creates synthetic time series with different frequencies (Fig. 1 top row). However, to achieve even lower frequencies with the same data we also re-arrange the original transient forcing based on the Greenland ice-sheet wide average annual air temperature. This changes the order of the 39 years in the record, but only entire years are moved. Note that this does not break the consistency between the atmospheric variables, or add energy or mass to the atmospheric system relative to the original natural forcing. Temporal continuity is only broken at the year break with arguably negligible consequences. The year break is in peak winter over Greenland with the entire ice-sheet being snow-covered, it can be assumed that non-linearity SMB feedback like the snow-ice-albedo play no roll. Note for a similar study at the southern hemisphere the temporal discontinuity at the calender year break may need to be re-evaluated.

The temperature-ordered forcing is created using different frequencies, with the example of one cycle in row two of figure 1. The entire forcing data in sequential arrangement is available in the appendix (fig. A1). All these time series have the same average forcing values, respectively the daily same climatology, but different temporal variability. They are obtained by ordering the 12 cycles of 39 years by the Greenland wide temperature from the coldest of the series to the warmest. Afterwards,

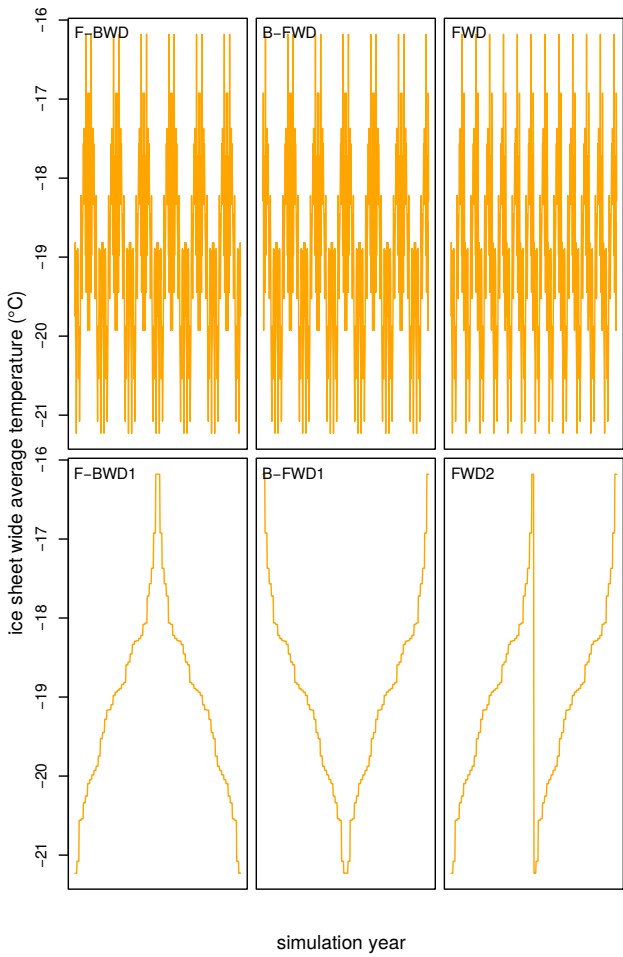

**Figure 1.** The forcing consists is based on 12 cycles of ERA-interim forcing. Each of them consist of the 12x39 years of ERA-interim. The first row shows the normal ERA-interim sequence (1979-2017) with different reoccurring patterns (2017-1979-2017x6,1979-2017-1979x6,1979-2017x12). In the second row there is an example of the temperature-ordered forcing with one cycle cold-warm-cold (F-BWD) and warm-cold-warm (B-FWD). Instead of looping back and forth from cold to warm the third colomn increase temperature and once the maximum/minimum is reached it starts over with the coldest/warmest forcing year again.

depending on the chosen frequency, we sample every $n$-th member of this series starting at the coldest/warmest year, where $n$ is the frequency and once the end of the series is reached we start over at the 2nd member sampling every $n^{th}$ member thereafter, this is repeated in total $n$ times for one time series.

These individual forcings allow us to investigate the sensitivity and feedback of the SMB to different inter-annual variability and, for example, extended warm periods.

## 2.3  Simulations

All simulations are spun up with 500 years of ERA-interim F-BWD to reach a stable firn cover. We then simulate the surface mass balance with the different forcing time series (sec. 2.2). The surface mass balance is calculated for five different total simulations with a duration of 78, 117, 156, 234 and 468 years, to mimic different temperature proxy resolutions

The first set of simulations use the unaltered transient climate forcing only reordered in time (FWD/BWD/F-BWD/F-BWD 1-12). The second set of simulations mixes climatological and transient forcing and the details will be explained in the results section. Lastly, we investigate the impact of the temporal precipitation distribution by simulating 468 years with the same monthly precipitation average but different sub-monthly frequencies.

## 3  Results

**Inter-annual variability - ordering** The average surface mass balance of the Greenland ice sheet is approximately 200 $\mathrm{kg\,m^{-2}\,yr^{-1}}$ independent of the ordering of the forcing years (Fig. 2). A figure with the SMB response of all simulations can be found in the appendix (fig. A2). The lowest SMB occurs if multiple warm years happen after each other displayed in the second row of figure 2. The memory effect of the firn cover to extended warm periods is rather low on an integrated level, though in the extreme case of only one cycle the SMB is slightly lower on the second cooling branch than the warming one. Within each frequency, BWD always shows the lowest SMB, because it starts with the warmest year and no protective firn cover can be built up first to reduce the amount of exposed ice. Note that we do not simulate changes in surface elevation, which could cause a significant positive feedback at multi-centennial time scales.

While the temporal order for the forcing years is of marginal influence (< 5% difference in SMB) over the entire ice-sheet, it is larger on a regional level. The variability mainly impacts the SMB around the equilibrium line, with a standard deviation of up to 500 $\mathrm{kg\,m^{-2}\,yr^{-1}}$ on the local scale (fig. 3). The standard deviation is also quite high in the northeast.

We also study additional simulation lengths of 78, 117, 156, and 234 years, corresponding to two, three, four, and six ERA-interim cycles. The general results are similar for shorter simulation periods (78, 117, 156, and 234 years instead of 468), though the difference between the simulations decreases, as with fewer ERA-interim cycles the duration of extended warm or cold periods decreases (not shown).

**Climatological forcing / Intra-annual variability** As the order of the inter-annual variability has a low impact, can we also simplify intra-annual variability further and use daily climatologies? We study the impact of the daily climatology for every variable individually and only for the B-FWD case. Two mixed data sets are created, one where all but one variable are held

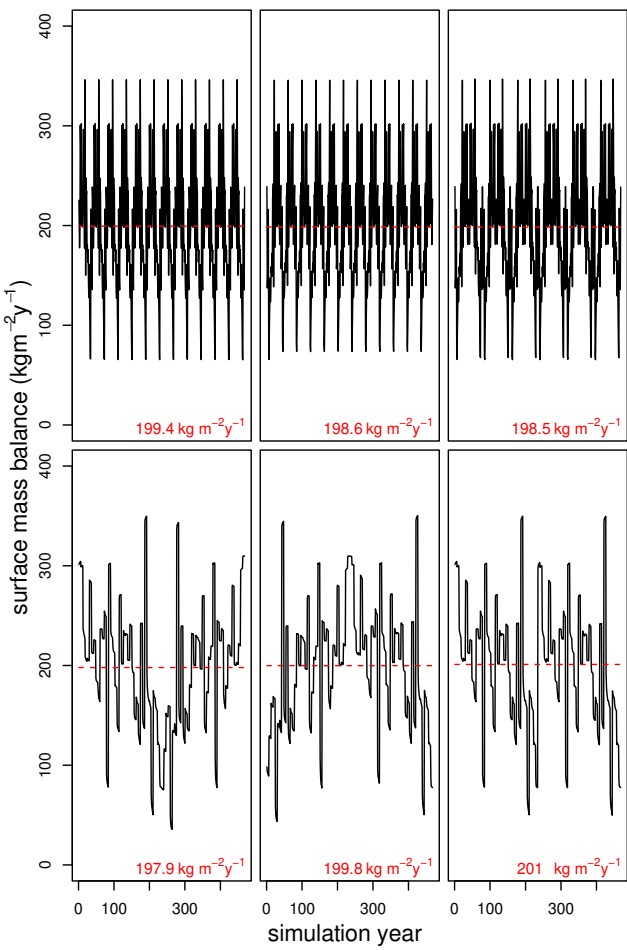

**Figure 2.** The SMB response of the Greenland ice-sheet to climate forcing for 12 cycles of ERA-interim forcing. Each box displays the annual surface mass balance over the entire simulation period of 468 years in black and the mean in red. The respective forcing is in the same order shown in figure 1, with F-BWD,B-FWD,FWD from left to right. The SMB for the natural forcing (row 1) varies by less than 0.5%. For the temperature-ordered case the lowest SMB is found for F-BWD which showes (lower left) has the longest period of consecutive warm years. The difference between the SMB with the different forcing is well below 5 %.

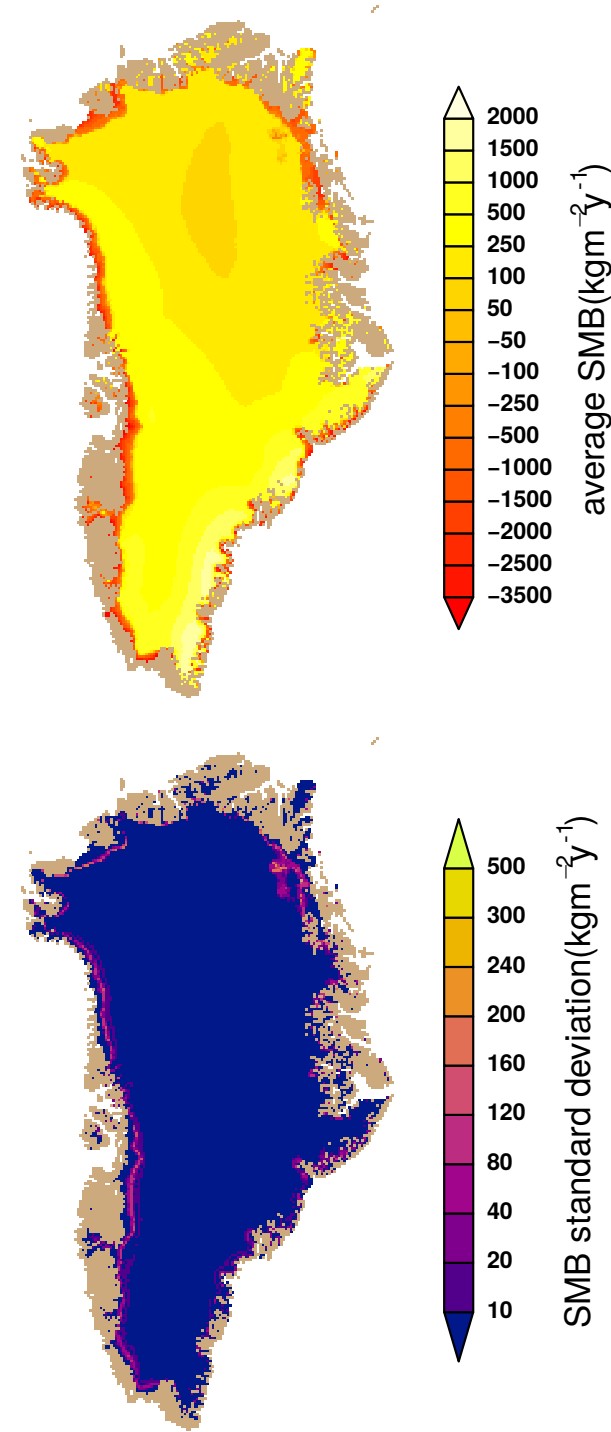

**Figure 3.** The average SMB and its standard deviation of the 28 simulations with different inter-annual variability order. The individual ensemble members all have the same climatology. The variation in the SMB is greatest around the equilibrium line and the northeast.

at their climatological averages, and vice versa, where only one variable uses the climatology. Based on the results from the previous section we select the B-FWD member as a representative for the transient forcing, as the other reordered time series yieled similar SMB values.

Forcing with the daily climatology leads to a drastic overestimation of the SMB by 40 % (274 $\mathrm{kg\ m^{-2}\ yr^{-1}}$ Fig. 4 a,b.). We further investigate this overestimation by studying the impact of the individual forcing variables: using a transient forcing for all but one variable, which is comprised of daily climatological averages (Fig. 4 right column), and the climatological forcing is mixed with one transient variable (Fig. 4 left column/ c-l). The SMB of these simulations exceed the transient forcing (4 b), meaning that using daily climatologies always lead to an SMB increase. This is no surprise due to the non-linearity of the SMB to energy input. There is a clear difference in the impact of the individual variables. While the climatological dew point increases the SMB by less than 2% (fig. 4 l), the radiation components increase the SMB by 5 % (fig. 4 h, j) with the long-wave radiation averaging having a larger impact. The transient dewpoint forcing also increases the SMB Average temperatures increase the SMB by 15% (fig. 4 d) and daily averages of precipitation increase the SMB by 30% (fig. 4 f). Vice versa the complementary effect is true for climatological forcing (fig. 4 c, e, g, i, k), with climatological forcing with transient precipitation showing the lowest SMB (fig. 4 e).

The small effect and low variability of the radiation components shows that using climatologies is justified in this case (fig. 4 g-j), as the inter-annual variability of Greenland wide radiation is relatively low anyway. Though it is still connected to a slight bias of 5% in the current climate. The turbulent latent heat flux has a relatively low impact on the Greenland wide SMB (Zolles and Born, 2021), which is in line with the low effect the dew point change has (fig. 4 k, l) although it is the only variable that increases the SMB for transient data. Humidity, given by the dewpoint as forcing, impacts the SMB via two aspects. First, with higher humidity the turbulent latent heat flux and its associated mass flux will either decrease sublimation (less mass loss) or increase condensation (more mass added). This effect is present over the entire ice-sheet. Secondly, the energy consumed for sublimation decreases or energy released during condensation increases with increasing humidity, leading to a heating up of the snowpack and potentially more melt. For the SMB this second effect only plays a role on decadal/short centenial timescales if melt occurs. Using climatologies of the dewpoint (a) compared to transient dewpoint (k) melt increases and SMB decreases at the margins, counteracted by a slight SMB increase in the center. The interior of Greenland experiences also a slight temperature decrease, i.e. increased sublimation more cooling. The absolute humidty is non-linear with respect to the dewpoint, with the averaging routine applied, the absolute humidity also decreases. With the interplay of energy and mass fluxes and the spatial and temporal variability of the latent heat flux, humidity does not continously increase the SMB in either state. Therefore, compared to all other variables the transient dewpoint increases the SMB in the climatological case (k) as well as climatological dewpoint in the transient case (l). A generalization of the effect of the dewpoint for all climate states is difficult due to the compensating effects over melt and non-melt areas.

While the biggest differences between the previous simulations were found around the equilibrium line (fig. 3), the largest difference between climatological and transient forced SMB simulations is found in the melting region of Greenland (fig. 5). Temperature has the second highest influence, which can be attributed mainly to the non-linearity of the SMB. However, the overestimation by climatological precipitation cannot be explained by the non-linearity, but by albedo. Using a daily

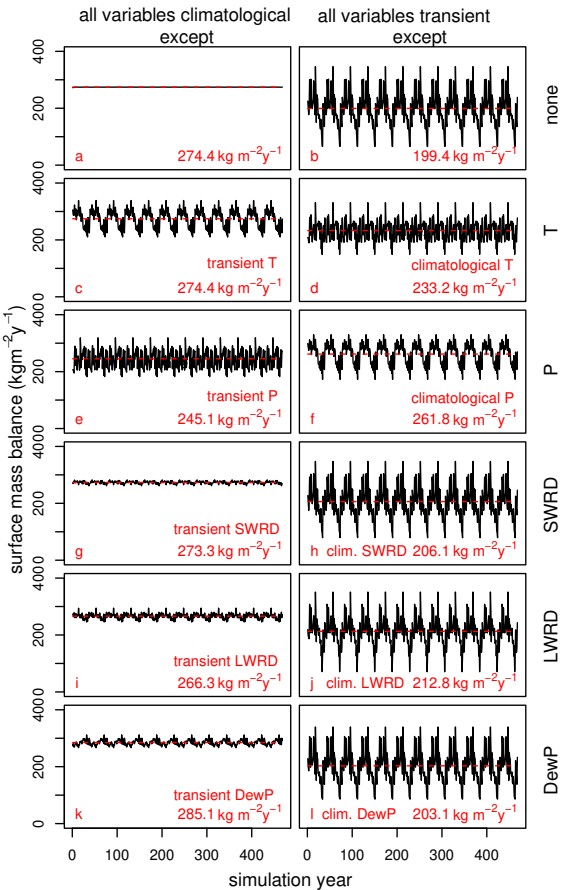

**Figure 4.** The Greenland wide integrated SMB with climatological and transient forcing. On the left side the model is forced with climatological (daily averages) forcing. In each row apart from the first, one forcing variable is transient instead. On the right, transient forcing mixed with one climatological variable is shown. The individual rows should be compared to eachother, rather than the columns for the discussion, as each row has one variable changing relative to the first row, while the right and left column are total inverses. The climatologically forced SMB model (a) overestimates the SMB relative to the fully transient case (b). If the model is forced with climatological variables except one, adding transient precipitation lowers the SMB the most (e). Vice versa climatological precipitation distorts and increases the "true" transient SMB the most (f). Transient dewpoint (k) increases the SMB. Climatological dew point, short-wave and long-wave radiation lead to slightly increased SMB (h, j, l).

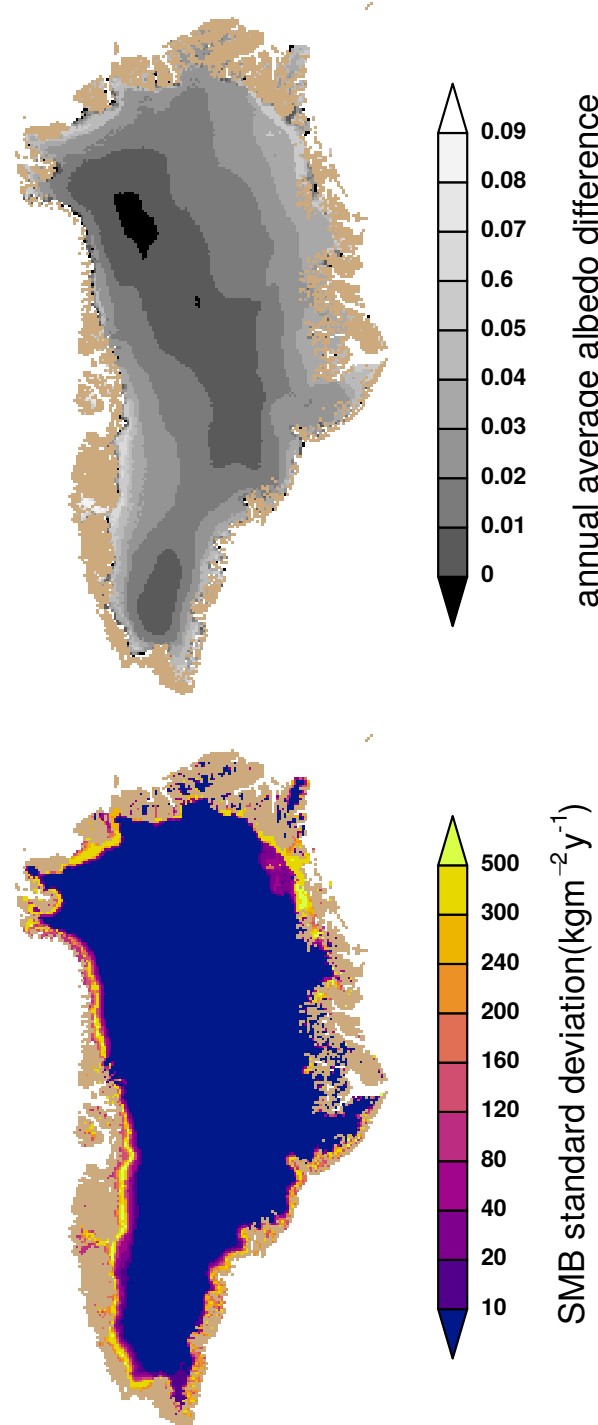

**Figure 5.** The standard deviation of the SMB for the transient and climatological mixed simulations (fig. 4) on the bottom, and the difference between the transient and the climatological forced surface albedo on the top. The largest standard deviation is in the melt region, due to an up to 0.1 larger annual average albedo.

climatology leads to small amounts of mostly snowfall every day leading to a surface albedo increase. The annual average albedo increase is up to 0.1 in the melt region of Greenland. The drastic effect of using daily climatologies of precipitation can be attributed to this albedo overestimation (Fig. 5).

**Can we emulate intra-annual variability of precipitation?** We have shown that BESSI overestimates the SMB drastically if daily climatologies of precipitation are used. A daily climatology is unrealistic as it has small amounts of snow fall every day. This does not agree with observations of highly event-based precipitation in the Atlantic region (Sodemann et al., 2008). We therefore calculate alternative temporal precipitation distributions by taking monthly averages with a sub-monthly distribution instead. Regular precipitation frequencies of 2, 4, 8, 15, and 30 days are tested as well as the sub-monthly distributions from each of the 39 ERA-interim years. For the ERA-interim based distributions the original daily time series $P_{day}$ is scaled to have the same monthly average:

$$P_{day} = P_{day}^t \cdot \frac{\overline{P_m}}{P_m^t} \quad \forall t \in [1979, 2017] \tag{1}$$

with $P_m^t$ as the monthly mean of the year t, and $\overline{P_m}$ the monthly climatological precipitation amount. This correction can be compared to the Delta Method for precipitation (Beyer et al., 2019). We obtain 39 possible precipitation time series, each with a different sub-monthly distribution of the precipitation analogous to the true precipitation of the specific year. Though the monthly sum of precipitation is the same for all the simulations the resulting distributions are quite different. April 2014 was a wet month, so for the resulting forcing it is scaled to have a lower monthly sum, but even then it has four days with precipitation of more than 10 $kg \ m^{-2}$ (fig. 6 b).

The simulated SMB depends on the chosen sub-monthly precipitation distribution (fig. 7). For regular precipitation the SMB decreases with decreasing precipitation frequency (255/233/200/154 and 87 kg m$^{-2}$ yr$^{-1}$ at precipitation every $2^{nd}/4^{th}/6^{th}/15^{th}$ and $30^{th}$ day). Independent of the forcing type of the other variables, reducing the frequency of precipitation from the extreme of the daily climatology decreases the SMB. This is also true for the sub-monthly distribution from the individual 39 ERA-interim years (fig. 7c, d). The precipitation-heavy years of the unaltered forcing are now showing lower SMBs than in the natural control (B-FWD) simulation. The regular precipitation distribution of 2-30 day precipitation frequency spans from 87 to 274 kg m$^{-2}$ yr$^{-1}$. As the actual precipitation amount over Greenland is spatially and temporally variable this approach may yield good results only by chance. Instead we superimpose the precipitation frequency of the ERA-interim time series on the monthly averages.

This (fig. 7d) reduces the mass balance by 30 kg m$^{-2}$ yr$^{-1}$ relative to the daily climatology (fig. 7b), which is much closer to the "true" value of the transient forcing (fig. 7a). The amplitude of this simulation's SMB time series is rather low as the same amount of precipitation falls every year, so it was investigated further. Instead of using different sub-monthly frequencies every year the distribution from each year ERA-interim year is taken as the forcing for the entire simulation period (as example 2009: fig. 7 f,g; the entire range is given in fig. 8). It spans from 224-253 kg m$^{-2}$ yr$^{-1}$. Using the sub-monthly precipitation distribution based on real analogs for the climatology reduces the SMB overestimation from 40% to 10-25%. A Greenland

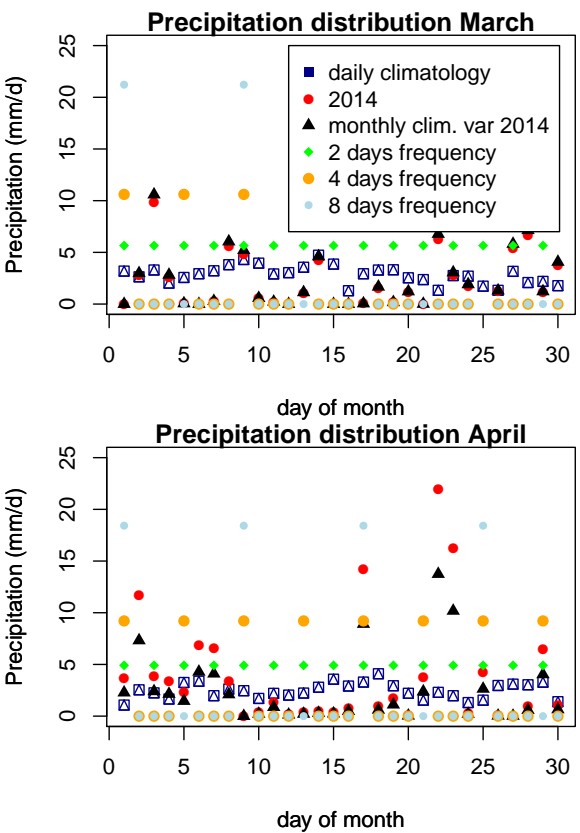

**Figure 6.** Sub-monthly precipitation distribution for March and April of different simulations. The same monthly precipitation is either distributed via daily climatologies (blue), monthly climatology with the sub-monthly distribution of, for example, 2014 (black) or with regular frequencies (green, orange, light blue). The red distribution is the true distribution for 2014 which is then scaled to the climatological average (black, eq. 1), as can be seen April 2014 was wetter than the average April of the ERA-interim period.

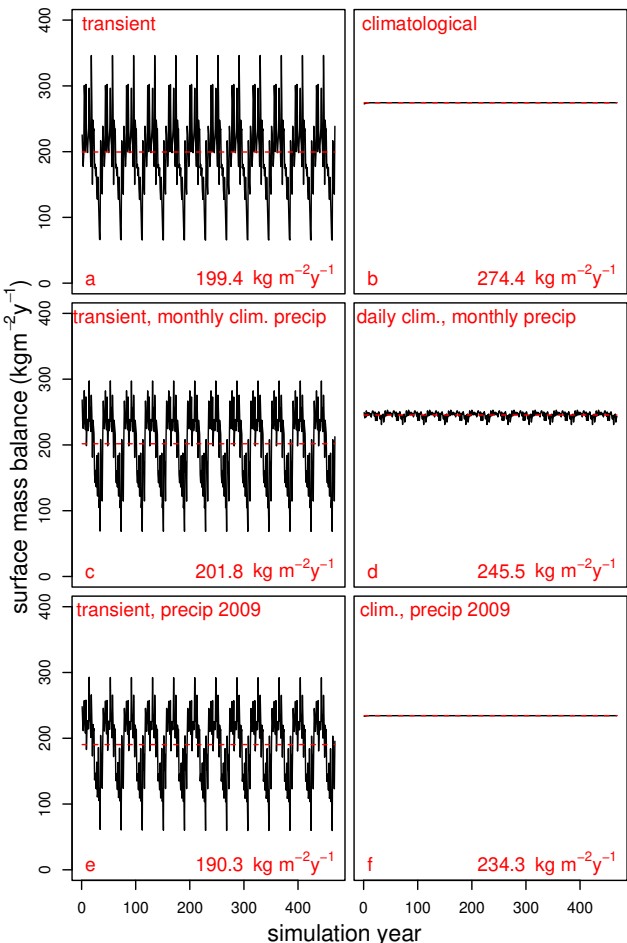

**Figure 7.** The Greenland-wide integrated SMB forced with different precipitation variability. The SMB time series is shown in black, with the average SMB as a red line. The SMB average value is shown in each panel. The transient B-FWD (a) and the full daily climatology (b) are shown again for direct comparison and are identical to Figure 4. The transient precipitation was scaled to have the same monthly average every year, with the sub-monthly frequency of the individual years, which is combined with either transient forcing (c) or daily climatological forcing (d) of the other variables. Similarly, we combine the sub-monthly precipitation distribution of one year, 2009, with transient (e) and the daily climatological (f) forcing of the other variables. 2009 was chosen as its monthly precipitation distribution is closed to the climatological average.

wide regular frequency may by chance show similar values as the transient simulation (8 days in this case), and 2-8 days fall with in the range of SMB values comparable to natural distributions.

The decrease in SMB is due to the non-linearity effect of the SMB, as in dry years earlier ice exposure triggers a feedback. Due to the non-linearity of the mass balance and albedo feedback, the range of these simulations is larger than the amplitude of

the single simulation (fig.7 d). Furthermore, part of the physical correlation between temperature and precipitation is removed, which further increases the differences.

## 4   Discussion

We study the impact of inter-annual climate variability on surface mass balance using a simple reordering of climate reanalysis data. The SMB shows a low dependency of 5% on the synthetic re-ordering frequencies in the forcing with the same clima-

tological average climate. In case of unknown inter-annual variability the use of a daily climatology forcing overestimates the SMB by 40 % due to the non-linearity of the SMB and albedo overestimation due to frequent small amounts of precipitation. Note that both these estimates do not include potential amplification by changes in ice elevation. We reduce this bias by imposing synthetic daily variations in the frequency of precipitation while keeping monthly averages unchanged. The overestimation of SMB is reduced but an uncertainty of 15% remains, depending on the chosen distribution is introduced.

Climate model simulations of the same time period vary in their inter-annual variability, they can very well represent the climatology, but not the order. We show that the effect of the order of the inter-annual variability is less than 5 %. This indicates that the memory effect of the Greenland wide integrated SMB to multiple warm or cold years is low enough to be modeled with climate forcing which may not have a realistic temporal variability. Even multiple subsequent warmer-than-average years over Greenland do not significantly lead to strong feedback without topographic adjustment. The last 40 years, using the

ERA-interim forcing, with its temperature trend (Hanna et al., 2021) can be considered an upper boundary for a "steady state climate". Similar results are to be expected for the improved 5-th generation ECMWF reanalysis, ERA-5 (Hersbach et al., 2023), but due to prior tuning of BESSI with ERA-iterim and ERA-iterim forced data, this study was conducted using these data sets. It is the same model setup apart from long-wave radiation down-scaling, which was used in the GrSMBMIP project (Fettweis et al., 2020a). We do not expect any qualitative changes for changing to ERA-5 data, though the extended length of

the ERA-5 dataset ranging back to 1940 would require less repetitive cycles of the same forcing to reach a simulation length of the order of 500 years.

The simulation lengths were 78,117, 156, 234 and 468 years, and even the extreme case of 12 consecutive years with the warmest temperature the average SMB only decreased by 3.5 %. If the climatology is known and the amplitude of the variability of the forcing data, the order does not really matter, despite the high inter-annual variability observed in line with

Van den Broeke et al. (2011). In case of climate simulations based on climatologies derived from proxies or other boundary conditions they likely are applicable for SMB simulations as long as the amplitude of the variability is known, even if there is a sub-resolution trend not visible in the proxy data. However, the effect is larger on a regional basis and around the equilibrium line the sensitivity towards this inter-annual variability increases. For the ERA-interim climate the northeast of Greenland with its sparse precipitation and large inter-annual variability in particular shows a standard deviation of up to $300 \, \mathrm{kg m^{-2} yr^{-1}}$.

If the inter-annual variability is not known, as is most often the case for the distant past or future, the forcing has to be based on climatologies. BESSI uses daily forcing data and is sensitive to daily precipitation, because it simulates snow aging and albedo decrease on a time scale of days. A small amount of snow-fall every day leads to an albedo overestimation as BESSI re-

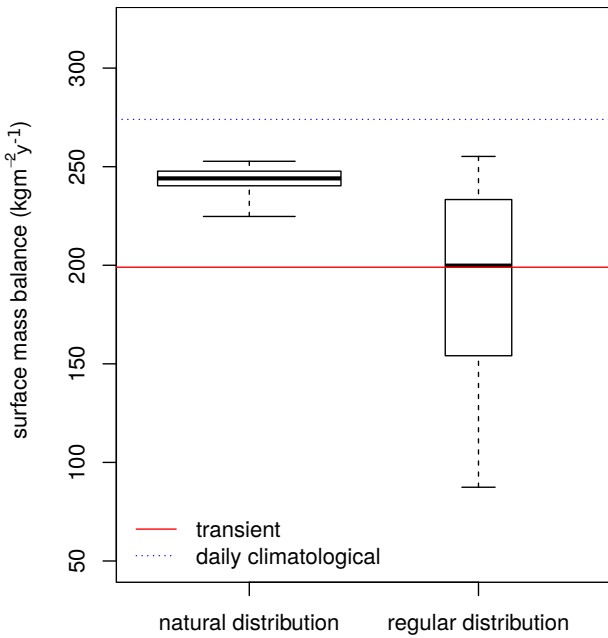

**Figure 8.** SMB averages for climate forcing with different precipitation variability based on the ERA-interim ensemble. In total 39 different sub-monthly natural precipitation distributions are shown on the left based on monthly averages distributed by the $39 \times 12$ sub-monthly distributions of each year 1979-2017. The SMB response to regular precipitation on the 2/4/6/15/30th day is on the right. The simulations are forced with the daily climatology of temperature, short and long-wave radiation, and dew point. The width of the boxes is relative to the size of the ensemble (39/5).

solves albedo adjustments on a daily basis. A possible solution is to parameterize the albedo routine differently for climatology and transient data. Alternatively, the precipitation climatology has to be calculated in a more physical reasonable way, as we

explore here. We show that monthly climatologies with a natural sub-monthly distribution reduce the SMB overestimation. In practice, there are multiple ways to define such a distribution: regular or stochastic frequencies for a region using normalized precipitation from reanalysis or climate simulation data. Either approach however, may be specific to the sampling period and not invariant in time, and multiple solution may exist. In the given case a regular frequency does not function well and may just fit by chance (Fig. 8 right box). The redistributing of the same precipitation amount at each grid point within a month can

change the SMB by 15% (fig. 8) even though all the distirbutions occured naturally over the ERA-interim time period (Fig. 8 left box). This is to be considered when selecting the fields for projections or reconstructions, purely based on scalar temperature and/or precipitation anomalies of a given field. The precipitation is quite variable in Greenland (Mosley-Thompson et al., 2005), but not only the total amount is important but also its temporal distribution, in particular in the melt region. There is no clear best representative of the precipitation variability among the individual years of the ERA-interim period.

Based on our findings we suggest that in the absence of full climate simulations with daily variability, temperature and precipitation anomalies are applied to a related climatology with sub-monthly frequency in precipitation. Still using climato-

logical forcing may be overestimating SMB, as it does for BESSI, due to the non-linearity of mass balance, which is in line with (Mikkelsen et al., 2018) who found a 13 % overestimation of the SMB if inter-annual temperature fluctuation is not considered. The choice of the representative precipitation distribution which is scaled is accompanied by an uncertainty of around 15%.

BESSI does not use sub-daily parameterizations for the daily cycle, which could reduce the effect of small amounts of snow falling every day and the accompanied albedo overestimation. Nevertheless, small amounts of precipitation every day are not physically reasonable for the region. The effect of the resulting albedo increase, even in the case of sub-daily parameterizations, overestimate the SMB and have to be considered in the snow models. BESSI showed a positive SMB bias in general relative to other snow-models, and we cannot state how big the mentioned effects are for the other SMB models (Fettweis et al., 2020b). BESSI was tuned against an RCM over Greenland, using the time series as well as the climatology in a multi-objective optimization approach. We therfore conclude it is not a result of overfitting to the tuning creating the high albedo sensitivity. Nevertheless, if BESSI is run with climatological data during the tuning the parameters are different. BESSI, and likely other simple energy balance models, should not be run and tuned for climtological/transient data and then run with the other. Using timely variable data (transient) for tuning is favorable due to the lower risk of overfitting. As transient data will not always be readily available for time periods of interest, we present a reasonable approach to create a superimposed precipitation forcing on climatological data to get better results, though an uncertainty of  15% remains.

We did not try to adjust climatological fields for temperature, or the other forcing variables. Due to the event based nature of the precipitation this has the biggest impact, but daily climatologies overestimate the SMB also due to the other variables too. The effect of the non-linearity of BESSI alone has been previously studied with the model (Born et al., 2019). We furthermore did not study the impact of precipitation distributions on the point scale.

## 5   Summary and Conclusions

A surface mass and energy balance model was run for 468 years with different climate forcing. Ever member of the ensemble of climate forcings has the same climatology in the five forcing variables, atmospheric temperature, precipitation, long and short-wave radiation, and humidity, with a variable temporal distribution. While different frequencies of climate variability have very little impact (< 5 %), using an average climate leads to a drastic overestimation (40 %) of the surface mass balance. This is mainly observed around the melt region of the Greenland ice sheet. The biggest contribution to this overestimation is the precipitation forcing ( 30 %), due to the resulting albedo increase. Averaging multiple years to obtain a climatology produces a data set with frequent light precipitation, and a high surface albedo due to the continuous presence of fresh snow. Small amounts of snowfall are not physically reasonable for a region with event based precipitation like Greenland.

To overcome the problem we calculated alternative precipitation climatologies to be used together with daily climatologies of the other variables. Monthly averages following a natural sub-monthly distribution lead to the smallest errors. However, there is a dependency on the chosen distribution. Using a regular frequency is not feasible as there is a large spatial dependency and empirical relations may change through time periods. We conclude that the surface mass balance model is best forced with

270 transient climate data. If daily climatologies with an altered precipitation forcing are used an overestimation of 15-25 % of the SMB should be assumed.

*Code availability.* The current BESSI model code is available on git-lab (https://git.app.uib.no/melt-team-bergen/bessi)

**Table A1.** The simulations done for this study.

| | Years | 78 y | 156 y | 234 y | 468 y | # of simulations |
|---|---|---|---|---|---|---|
| ERA-interim | B-FWD | 1 | 1 | 1 | 1 | 4 |
| | F-BWD | 1 | 1 | 1 | 1 | 4 |
| | FWD | 1 | 1 | 1 | 1 | 4 |
| | BWD | 1 | 1 | 1 | 1 | 4 |
| temperature-ordered, 6 frequencies | B-FWD | 6 | 6 | 6 | 6 | 24 |
| | F-BWD | 6 | 6 | 6 | 6 | 24 |
| | FWD | 6 | 6 | 6 | 6 | 24 |
| | BWD | 6 | 6 | 6 | 6 | 24 |
| Point wise temperature-ordered, 6 frequencies | B-FWD | | | | 6 | 6 |
| | F-BWD | | | | 6 | 6 |
| | FWD | | | | 6 | 6 |
| | BWD | | | | 6 | 6 |
| Daily climatological forcing | | 1 | 1 | 1 | 1 | 4 |
| Mixed forcing, 1 climatological variable | T, P, SW, LW, DewP | | | | 5 | 5 |
| Mixed forcing, 1 transient variable | T, P, SW, LW, DewP | | | | 5 | 5 |
| Other precipitation climatologies | Sub-monthly natural | | | | 39 | 39 |
| | Regular 2,4,8,15,30 | | | | 5 | 5 |
| | mixed | | | | 3 | 3 |
| | | | | | | 197 |

*Author contributions.* TZ conducted the model tuning and ensemble simulations, the data analysis and wrote the main part of the manuscript. AB contributed to the study design, and the manuscript.

*Competing interests.* The authors declare that they have no conflict of interest.

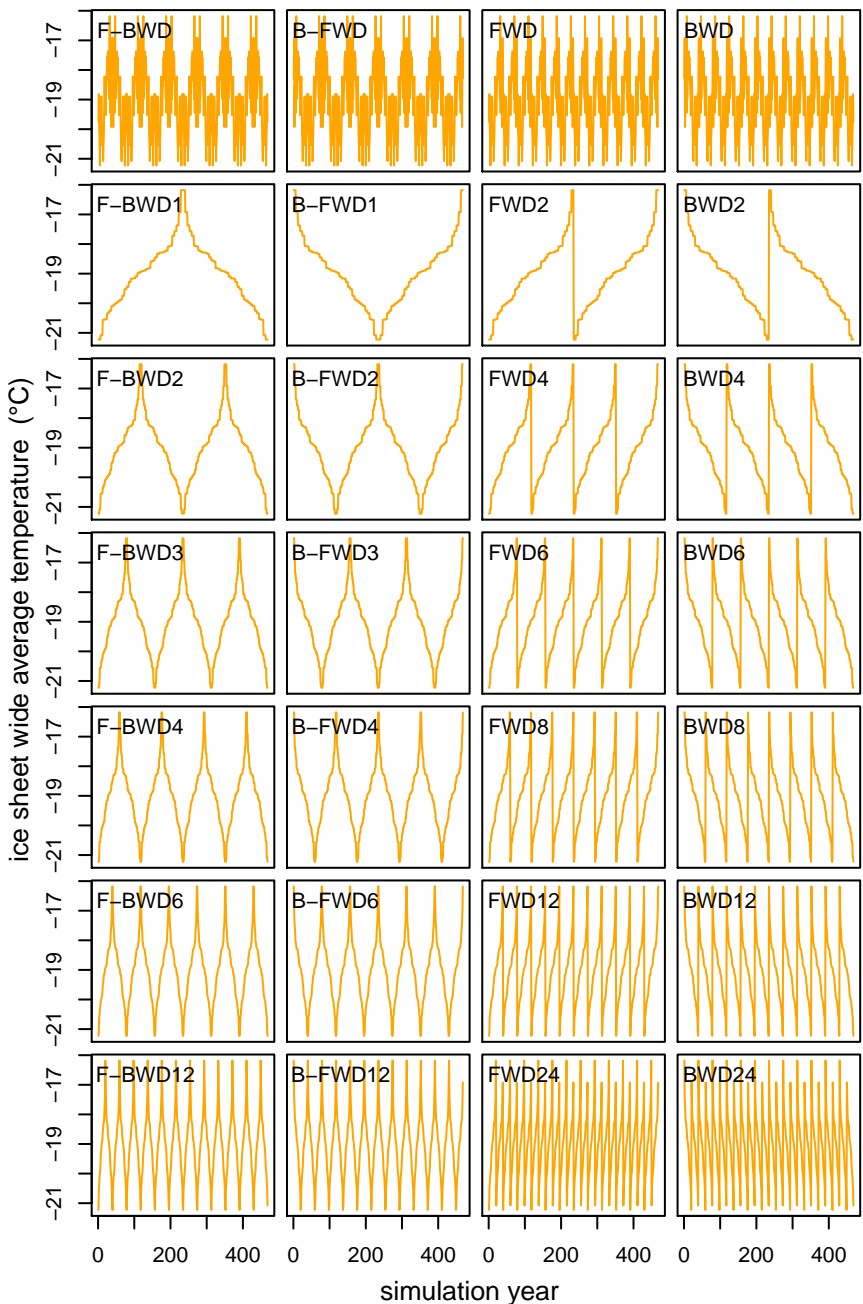

**Figure A1.** 28 different temperature time series based on 12 cycles of ERA-interim forcing. Each of them consist of the 12x39 years of ERA-interim which are ordered by temperature with different reoccurring frequencies. The first row shows the normal ERA-interim sequence (1979-2017) with different reoccurring patterns (2017-1979-2017x6,1979-2017-1979x6,1979-2017x12,2017-1979x12). Rows three to six show the temperature-ordered sequence with increasing frequencies, with row one starting cold (F-BWD) and row two starting warm (B-FWD). Instead of looping back and forth from cold to warm the last two rows (orange) only increase/decrease in temperature and once the maximum/minimum is reached it starts over with the coldest/warmest forcing year again.

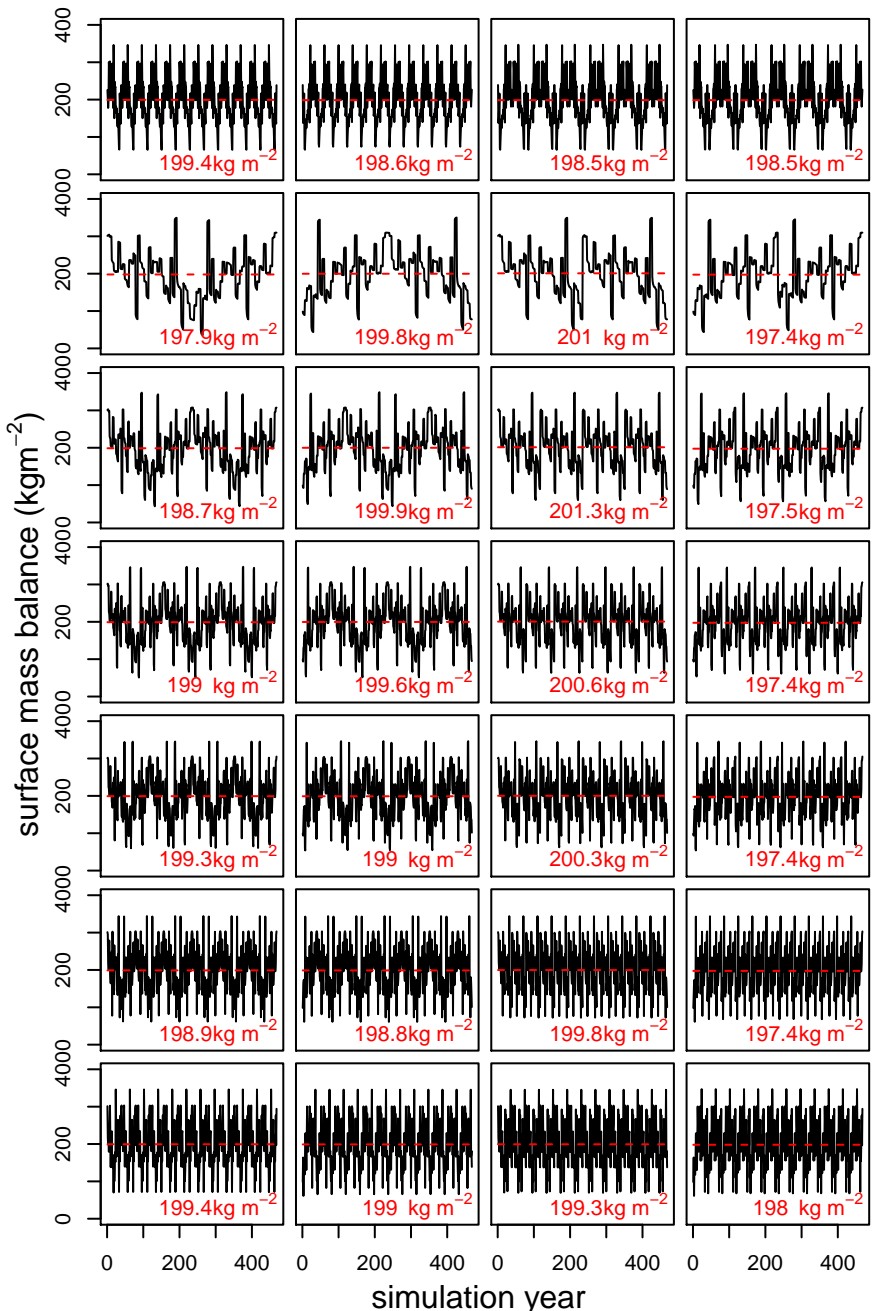

**Figure A2.** The SMB response of the Greenland ice-sheet to climate forcing with different inter-annual variability. Each box displays the annual surface mass balance over the entire simulation period of 468 years in black and the mean in red. The respective forcing is in the same order shown in figure 1, with F-BWD,B-FWD,FWD,BWD from left to right. The difference between the SMB with the different forcing is below 5 %.

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
