# Peer review of "How does a change in climate variability impact the Greenland ice-sheet surface mass balance?"

_The Cryosphere, 2021_

## Author Comment (AC1)

Answer to Anonymous Reviewer 1.

We would like to thank the reviewer for his thorough assessment of our work and the feedback. We will accompany the adjusted changes and will provide some comments below. The final individual remarks to every point will be published together with the revisions.

This study investigates the impact of temporal variability in climatic forcing to drive a snowpack model over Greenland. The idea is that for many time periods of interest, climatic datasets of daily or higher resolution are not available, but are needed to drive snowpack models. Therefore different strategies may be needed to obtain consistent results using this sparse forcing with a model that is tuned based on high-resolution climatic data. It is a valuable study that should be published with only minor revisions.

The experimental setup is interesting, and serves to demonstrate the importance of accounting for variability in the climatic forcing. However, I get the impression that a lot of details are provided for the different ordering of years (Figs. 1 & 2, 28 panels each), when in fact, it is determined that the shuffling of the years does very little to change the esimated SMB in the end. I would recommend relegating all of these panels to an Appendix, and rather show two representative cases of the forcing and resulting SMB in one figure. This allows you to make the point, and if the reader is interested they can check the other cases in the Appendix. But importantly, then it brings the focus more to your main point, which is the intra-annual variability.

Thank you for this remark. We will try to streamline and shorten the section about the ordering of the years, nevertheless, this was hardly ever investigated for different models and it is a key part of the study that we think is necessary before looking in the removal of intra-annual variability by using climatologies. Fig 1&2 will be adjusted to accompany fewer panels in the main manuscript and potentially moving the rest to an appendix.

With regards to the intra-annual forcing, the findings here are quite valuable. It is clear that if a model is tuned with historical daily input fields, forcing it by climatological averages of daily input fields can result in strong biases in the simulated SMB. The study nicely diagnoses that precip is the key factor here, while climatological averages of other variables do not increase the bias much. The proposed method to reduce this bias is also valuable and nicely tested.

However, I am less convinced by the idea that imposing a little bit of precipitation each day is problematic. By using daily forcing, the model is already being driven by forcing that is "not realistic", since it does not capture some of the strongest variability in the fields - namely the diurnal cycle. And yet, it can be tuned to do a good job against an RCM.

We would like to thank the reviewer for the appreciation of our work. Within the given albedo routines, we found daily precipitation to be an issue. It is possible that the effects are reduced for models that are tuned for climatological data. We accept that for the diurnal cycle qualitatively similar results as for the inter-annual variability on different temporal scales are to be expected. Resolving the diurnal cycle – though desirable comprises with our wish for numerical efficiency. Additionally, BESSI uses relatively large boxes of around one meter at the top, which would dampen any diurnal cycle imprint on the snowpack anyway.

My suspicion is that if BESSI were tuned against the climatological SMB of RACMO while driven by climatological-average variables, it would still be able to produce a reasonable estimate of climatological SMB. Based on the analysis given here, one could guess that the optimal albedo

parameters would change to reduce the sensitivity of the model to precip. And then, in principle, it would be ok to use climatological variables from other time periods. Would it be possible for the authors to test this easily? I would not say it is a requirement for publication, but at a minimum, it would be good to include some discussion of this possibility and its implications.

In principle we can tune BESSI for the climatological mass balance of RACMO for example. Though this is part of the multi-objective optimization conducted, in this study we did not tune against only the climatological mass balance. The albedo parameters are lower in case of tuning for climatological SMB, but the effect is less than expected as nevertheless within our tuning we only allow for a single albedo value for fresh snow over the entire ice sheet. We will look into detail if it is possible to also run the model with a calibration based on climatology.

Minor comments:
L7: "However, using daily averages as forcing ..." <= This could use some clarification. What kind of data were you using before, that were not daily averages? I.e., what are you contrasting to here?
L43: weather => whether
L45: Global Circulation => General Circulation
L48: prior => previously
L70: We use => As forcing, we use
L83: (rows) => (rows in Fig. 1)
L134: where found => were found
L156: I note here that the SMB changes drastically when a frequency of 30 days is imposed - SMB goes down to 87 kg/m2/yr from 255 kg/m2/yr, so it seems you can get any SMB you want bracketing the 'right' value using historical forcing. So, it is not clear why the bias remains at "10-25%" (L165) using this approach.
We will make this clear as 10-25% is a result of the natural based forcing not on the regular frequencies of 2/4/8/15/30 days.
L166: decreases with precipitation frequency => decreases with decreasing precipitation frequency [right?]
yes
L195: physical more => physically
Fig. 8: The meaning of this figure is not really clear to me. As I'm not really sure what is being shown, I cannot offer suggestions for improvement.
We will improve the figure and figure description
L212: physical not reasonable => not physically reasonable

---

## Author Comment (AC2)

Answer to Anonymous Referee 2

We would like to thank the referee for his comments and feedback on the study. Please find the individual remarks below.

This is an original and interesting paper on the effect of climate variability at different timescales (daily to monthly) on the Greenland Ice Sheet surface mass balance, and has potentially important implications for the way that climate forcing data should be used in SMB simulations, as there can be quite large differences (up to several tens of percent) depending on the type and time resolution of forcing data used. I have a few minor comments for the authors' consideration, following which I recommend publication.

ERA-5 data are available back to 1950 and are based on a superior model. Why was the older and shorter ERA-I dataset used?

We are currently using ERA-5. However this study is based on simulations and a calibration of BESSI that stems from a previous paper and therefore uses ERA-interim. We do not expect our results to critically depend on this choice.

p.3, line 82 "Temporal continuity is only broken at the year break with arguably negligible consequences". Has this been checked for days near the beginning or end of the year, as weather conditions may be very inconsistent with different years spliced together?

The temporal continuity is broken due to the set-up. It has no effect on the numerical stability on the snow model, as also SMB response is limited we see negligible consequences. This may be a bigger issue if a similar study is to be conducted for the southern hemisphere as more feedbacks are active during the summer with for example the ice-albedo feedback.

p.8, l.133 "...which is in line with the low effect the dew point change has (fig. 4 k, l)" - itlooks like there is quite a large change in the means of dew point relative to other climate variables, so can this point be clarified?

The dewpoint has the least impact if only its climatological average value is used to force BESSI (fig 4 l). We will clarify this in the revised version of our manuscript.

Re. Figure 4 caption comment "If transient variables are taken individually the precipitation lowers SMB the most", I don't fully follow this. To me the means look quite close for panels e & f for precipitation. Other climate variables have their respective climatological and transient forcings affecting their means by typically greater amounts.
Also, the labels "all climatological except" and "all transient except" at the top of Fig. 4 seem unclear and should be clarified.

We will improve the caption and labeling of figure 4. We will furthermore elaborate which graphs should be compared as the statement about precipitation is not from comparing the left and the right side, but rather the top with its respective line (a-e and b-f) in case of precipitation for example. In either case e and f deviate the most from a/b of all the individual variables.

---

## Author Response (AR1)

Dear editorial team,

Based on the reviews by the anonymous reviewers we adjusted our manuscript.
- Figure 1 and 2 were moved to the appendix and only selected panels are shown in the main text
- The figure captions were improved
- The abstract was reworked
- We expanded the setup as well as the discussion about tuning process and potential deficiencies of BESSI upon tuning to transient data
- All figures had the unit changed for the SMB from $kgm^{-2}$ to $kgm^{-2}y^{-1}$
- We added a short section about the temporal continuity at the year break
- We implemented all other suggested changes by the reviewers

All the changes are directly visible in the markup file provided.

Best regards
Tobias Zolles

---

## Author Response (AR2)

Answer to the anonymous referee 2

We would like to thank the referee for providing feedback on our study a second time. We add the remarks from the reviewer from the initial review in this document too:

"There is no proper response to reviewers document. The points in my original review have not been adequately addressed (e.g. use of ERA5-v.ERA-I reanalysis dataset) or those that have are unclear (labels on Fig. 4; large dew-point changes shown in Fig. 4). I suggest a revision is necessary which addresses point by point all the points in the original reviews. Until this is done, I am unable to recommend acceptance."

We realize that our reply in the first round of revisions was too brief. We therefore include the original comments here again, along with a more detailed explanation.

"ERA-5 data are available back to 1950 and are based on a superior model. Why was the older and shorter ERA-I dataset used?"

The primary objective of our study is to quantify the effects of using either transient or climatologically averaged data. All comparisons are done within this consistent framework that does not directly depend on the overall realism of the simulations. Hence, we do not expect a qualitative difference between the two reanalysis data sets. It is possible that the superior data set results in a slightly better simulation of the average SMB over Greenland, not least because of its higher resolution and a better representation of the ablation region. In addition, significant differences of ERA5, e.g., about 1K colder temperatures over Greenland when compared to ERA-interim (Krebs-Kanzow et al., 2023) would lead to a different calibration of BESSI. While it is conceivable, for a nonlinear system, that this change in model parameters does not only change the average state but also its sensitivity to variations, we deem this possibility unlikely and outside the scope of this present study. Lastly, our study uses a model version of BESSI that was calibrated using ERA-interim and data from the RACMO2 model that also used ERA-interim as input data. It is the same model version that was used in the GrSMBMIP project (Fettweis et al., 2020). These considerations added to our decision to continue using the ERA-interim data set for this study.

We added the following paragraph to the manuscript. A version of this text was also included in our first revision, but we failed to highlight this in our point-by-point reply. This new text can be found in the revised manuscript on page 14.
*"The last 40 years, using the ERA-interim forcing, with its temperature trend (Hanna et al., 2021) can be considered an upper boundary for a "steady state climate". Similar results are to be expected for the improved 5-th generation ECMWF reanalysis, ERA-5 (Hersbach et al., 2023), but due to prior tuning of BESSI with ERA-iterim and ERA-iterim forced data, this study was conducted using these data sets. It is the same model setup apart from long-wave radiation down-scaling, which was used in the GrSMBMIP project \citep{SMBIP2020}. We do not expect any qualitative changes for changing to ERA-5 data, though the extended length of the ERA-5 dataset ranging back to 1940 would require less repetitive cycles of the same forcing to reach a simulation length of the order of 500 years."*

Figure 4 caption comment "If transient variables are taken individually the precipitation lowers SMB the most", I don't fully follow this. To me the means look quite close for panels e & f for precipitation.

Other climate variables have their respective climatological and transient forcings affecting their means by typically greater amounts.
Also, the labels "all climatological except" and "all transient except" at the top of Fig. 4 seem unclear and should be clarified.

In addition to adjusting the caption in the first revision, we changed the labels to include more information within each panel.  This now includes directly highlighting the parameter to focus on. after consulting with several colleagues from within and outside our field, we decided to keep the original structure using 2x6 panels structure, but modified the labels for each column and row.

p.8, l.133 "...which is in line with the low effect the dew point change has (fig. 4 k, l)" – it looks like there is quite a large change in the means of dew point relative to other climate variables, so can this point be clarified?

We furthermore investigated the effect of the transient dew-point. Which has a rather large effect on the otherwise climatological forced SMB (fig. 4k). We analyzed total SMB, melt, and the turbulent latent heat flux over Greenland for this, comparing the simulations for 4a and 4k. The results are added to the results section Climatological forcing / Intra-annual variability around line 130-150. As the effect of the dew-point on the SMB is two fold in case of melting conditions and dry conditions, it varies over space and time. While in the melting region a lower dew-point will result in an increase in SMB (cooling due to more sublimation), it will result in a decrease (mass loss) in the dry region, to said sublimation. Drawing conclusions out of this dewpoint effect for different climate states is therefore more difficult than for the other parameters and we therefore did not focus on it in the initial submission.

Answer to Robinson Alexander

We would like to thank the reviewer for the second assessment of our work:

The authors have thoroughly revised the manuscript, and I believe it has improved significantly. I would offer only the following minor suggestions for improvement before publication:

L16: sea level rise => sea-level rise

L17: becomes the dominant component => becomes dominant

L70: Consider deleting this line and instead simply stating something like "as explained below", since here you have not yet defined the terminology F-BWD etc. You could put this specific example below after defining the terms.

L80: "less" here is not clear, can you rephrase somewhat?

L82: temperature ordered forcing => temperature-ordered forcing

L121: The daily climatology => Forcing with the daily climatology

L123: which comprises of => which is comprised of

L125: that daily climatologies => that using daily climatologies

L140: effect of => effect of using

L154: monthly some => monthly sum

L154: even than => even then

L166: this simulations => this simulation's

L167: it was => so it was

L190: steady state climate => a steady-state climate

L190: climate, similar => climate. Similar

L192: interim.. => interim.

L194: inter annual => inter-annual

L203: bases => basis

L204: a physical more reasonable way => a more physically reasonable way

L206: ways how to => ways to

L223-224: Not a complete sentence, please revise.

L224: we cannot => and we cannot

L230: As not in all cases transient data will be easily available => As transient will not always be readily available

L238: "They all share the same climatology ..." <= This sentence is unclear, please revise.

L247: Though => However

L250: transient climate => transient climate data

Best regards,
Alex"

We applied all the changes listed above and thank the reviewer again for their work.